# Characterization of Male Flower Induction by Silver Thiosulfate Foliar Spray in Female Cannabis at the Middle Reproductive Stage for Breeding

**DOI:** 10.3390/plants13172429

**Published:** 2024-08-30

**Authors:** Juyoung Kim, Dong-Gun Kim, Woon Ji Kim, Ye-Jin Lee, Seung Hyeon Lee, Jaihyunk Ryu, Jae Hoon Kim, Sang Hoon Kim

**Affiliations:** Advanced Radiation Technology Institute, Korea Atomic Energy Research Institute, 29 Geumgu-gil, Jeongeup-si 56212, Jeollabuk-do, Republic of Korea; jykim83@kaeri.re.kr (J.K.); dgkim@kaeri.re.kr (D.-G.K.); wjkim0101@kaeri.re.kr (W.J.K.); yjinlee@kaeri.re.kr (Y.-J.L.); ishyeon@kaeri.re.kr (S.H.L.); jhryu@kaeri.re.kr (J.R.); jaehun@kaeri.re.kr (J.H.K.)

**Keywords:** cannabis plants, silver thiosulfate (STS), male flower induction, pollen viability, cannabis breeding

## Abstract

Cannabis (*Cannabis sativa*) is a versatile crop belonging to the Cannabaceae family, and is dioecious, typically with separate male and female plants. The flowers of female plants, especially the trichomes, accumulate relatively higher contents of cannabinoids compared with those of male plants. For this reason, to obtain seeds that are genetically female, it is desirable to induce the development of male flowers on a female plant that produces genetically female haploid gametes. Silver thiosulfate (STS) is a highly effective chemical for male flower induction. We investigated male flower induction in three commercial cultivars of female cannabis (Spectrum303, SuperwomanS1, and CBGambit) regarding the treatment frequency, stage of application, and concentration of STS applied as a foliar spray. All three cultivars showed adequate induction of male flowers in response to 1.5 mM STS applied at the early reproductive stage. In particular, SuperwomanS1 was most highly responsive to induction of male flowers, even when treated with 0.3 mM STS at the early reproductive stage. Treatment with three applications of STS was more effective compared with a single application, but a single application of 1.5 mM STS at the early reproductive stage was sufficient for male flower induction. A single STS application during the middle stage of reproductive growth was inadequate for induction of male flowers. However, 6 weeks after three applications of STS, CBGambit exhibited approximately 54% male flower induction at 0.3 mM STS, Spectrum303 showed approximately 56% induction at 3 mM STS, and SuperwomanS1 yielded approximately 26% induction at 1.5 mM (expressed as percentage of total number of individuals with the induced male flowers). Pollen stainability tests using KI-I_2_ solution and Alexander’s staining showed high pollen viability with over 65% at different single STS concentrations, indicating that pollen grains induced by STS have sufficient viability for the self-pollination. This study demonstrated that different cultivars of cannabis respond diversely to different STS concentrations and highlighted the potential benefits of three STS applications during the middle reproductive stage for cannabis breeding.

## 1. Introduction

Cannabis (*Cannabis sativa*) is a flowering plant classified in the Cannabaceae family. Historically, the species was cultivated in the temperate zone of western and central Asia, extending to eastern Asia and Europe. Cannabis has mainly been used as a source of textile fiber or natural therapeutic agents, especially in ancient China for more than 6000 years [1,2]. By employing advanced chemical analysis, more than 100 terpenophenolic compounds, which are regarded as natural therapeutic agents, have been categorized as cannabinoids. The most distinctive compounds synthesized in cannabis are two cannabinoids, delta-9 tetrahydrocannabinolic acid (THCA, a narcotic) and cannabidiolic acid (CBDA). Based on these traits, cannabis has been colloquially referred to as hemp and marijuana, which are sometimes used interchangeably. However, nowadays, these two terms are more precisely defined. Hemp refers to specific cultivars or varieties of cannabis that are primarily grown for industrial purposes such as fiber, oilseeds, and other products (not intended for use as a euphoric drug). Marijuana refers to the dried flowers, leaves, and stems of specific cultivars of cannabis that are mainly grown for recreational or medicinal drugs for their psychoactive effects. Therefore, in several countries, cannabis is classified as hemp if it contains less than 0.3% THC (international standard, per dry weight, with some variation according to national policy), and marijuana if it contains a higher THC content than that of hemp or more than 3% THC [3,4].

Cannabis is an incredibly versatile and multifunctional crop. In addition to the known uses of cannabis, recent reviews on cannabis have highlighted specific industrial uses of various organs such as the seeds, flowers, leaves, stems, and roots. These organs can be used for many specific purposes, including the production of paper, bioplastics, biofuels, fabrics, construction materials, beverages, cosmetics, agricultural products, biopesticides, medicine, and food, as well as for phytoremediation [5]. Among the various organs of cannabis, female flowers are traditionally used for medical and recreational inhalation. Female flowers are particularly important in cannabis cultivation because they accumulate the highest contents of cannabinoids, such as CBD and THC, compared with other tissues, specifically in trichomes of female flowers.

With an increase in the demand for cannabis, the breeding of cannabis cultivars for specific end uses has intensified. Regarding the use of hemp in the fiber industries, the focus is on using polysaccharides extracted from the stems and leaves to create biomaterials. Therefore, breeding has focused on increasing biomass. In addition, the seeds can be used to extract an edible oil; hence, there are efforts to improve the composition of unsaturated fatty acids and various minerals and beneficial bioactive compounds in the seeds [6]. For example, Schultz [7] compared the components of approximately 20 seed or fiber cannabis cultivars grown in Australia. With respect to medicinal uses, the focus is on cannabinoids, and specifically high CBD and low THC contents. Many studies have investigated medicinal application of cannabinoids, such as reduction of chronic pain, relief of symptoms of inflammatory diseases, anorexia, and AIDS [8,9]. Therefore, breeding research aims to increase or moderate the accumulation of these substances in relation to treatment of the aforementioned medical conditions [10,11]. In order to achieve the manipulation of cannabinoids, the need to continuously maintain female plants has arisen, since growers prefer cultivating female plants for medicinal purposes because they tend to accumulate higher contents of cannabinoids than male plants [12,13].

Cannabis is a diploid species with 2*n* = 20 chromosomes that include two sex chromosomes [14]. Sex expression in cannabis is influenced by various genetic factors, hormones, and the sex chromosomes [15,16,17]. This can result in the appearance of monoecious or hermaphrodite plants, which have both male and female reproductive organs on one plant [16,18]. Using monoecious or hermaphrodite properties for seed production, female flowers of a female plant can produce seeds with 100% XX chromosomes when pollinated with pollen from male flowers on the same female plant, whereas normal cannabis plants can produce seeds with an XX:XY = 1:1 chromosome ratio [19]. Thus, the cultivation strategy for the production of higher cannabinoid contents involves the removal of male plants at the seedling stage or before flowering. However, the use of this strategy to distinguish genders is tedious. To overcome this hurdle, producing feminized seed has been devised by induction of male flowers by a female hemp plant using hormones and chemicals [16]. Gibberellin A3 is a hormone that can induce male flowers in cannabis, but fewer male flowers are induced than with the chemicals silver thiosulfate (STS) or sodium nitrate [20]. Sodium nitrate and STS are currently the most commonly used chemicals for induction of male flowering, although the effectiveness for male flower induction differs depending on the concentration applied. The aforementioned studies emphasize the need for a large amount or frequent application of chemicals to induce many male flowers for medical and industrial cannabis uses. However, to produce an appropriate number of seeds from a single cannabis plant for breeding purposes, induction of less than 50% male flowers may be sufficient. Given that analysis of the cannabinoid contents in separate groups for breeding should be performed during the mid- or late-flowering period, after conducting the analysis, some of the female flowers should be converted into male flowers for self-pollination to obtain seeds. Therefore, it is necessary to investigate whether it is possible to induce male flowers in female cannabis plants using fewer or lower doses of sex-converting chemicals, or to induce male flowers at a mid-flowering stage so as to isolate desired breeding varieties.

The viability of pollen can be assessed by staining methods to check its vitality or by examining pollen germination [21]. Representative examples of staining methods include KI-I_2_ solution and Alexander’s staining [22,23]. Viability tests of cannabis pollen have primarily been conducted using in vitro pollen germination assay on pollen grains obtained from male plants [24,25,26]. Studies have also reported that pollen from male flowers induced by chemical inducers, such as STS, is viable both in vivo and in vitro [20,27]. However, studies on the changes in pollen viability depending on the concentration of STS are still limited.

To enable self-pollination through male flower induction in female cannabis plants, an appropriate degree of sex conversion is required. Moreover, it is necessary to determine the appropriate chemical concentration and application frequency in accordance with the growth stage. In this study, we investigated the efficiency of male flower induction in female cannabis plants treated with STS at different concentrations and application frequencies at the early or middle reproductive stages. We also assessed the stainability of pollen from the induced male flowers and seed production as an indicator of pollen viability.

## 2. Materials and Methods

### 2.1. Plant Materials and Growth Conditions

The three cannabis cultivars used in this study were all feminized female plants: Spectrum303 (received from Jeonbuk National University), SuperwomanS1, and CBGambit (Trilogene, Longmont, CO, USA). Plants were propagated from cuttings after seed germination to produce a sufficient number of individuals. Experiments were performed from 5 September 2022 to 27 February 2023 except for the pollen stainability test. During this period, all plants were grown in a glass house under controlled environmental conditions. The cuttings were prepared by excising a shoot approximately 10 cm in length and inserting the base into the soil in an 8 × 9 cell universal seedling tray. The temperature and relative humidity for rooting were maintained for 2 weeks at 26–27 °C and more than 80%, respectively, and the light intensity was approximately 100 photosynthetic photon flux density (PPFD, μmol m^−2^ s^−1^). For the following 1 week, the temperature was unchanged, but the humidity was maintained at 30–50%. The period from excision to rooting took approximately 3 weeks. After transplanting the rooted cuttings into individual pots (diameter 12 cm, height 10 cm), the vegetative growth conditions were maintained at a temperature of at least 24 °C and relative humidity of 30–50% with 16 h/8 h (light/dark) photoperiod. The plants were grown in a smart farm in a glass house at 150 PPFD until they reached approximately 25 cm height (when the leaves reached the fifth alternate phyllotaxis) under long days. The plants that had completed their vegetative growth were transferred to a short-day growth room to induce flower formation and were treated with STS. For reproductive growth, natural light was provided for 12 h ± 30 min. The temperature was maintained at more than 20 °C with relative humidity of 27–50%. The plants were irrigated regularly, and no fertilizer was applied during short day treatment until seeds were harvested.

### 2.2. Phenological Growth Stage of Female Cannabis Used in This Study

We divided reproductive development into three stages, namely, the early stage (from the first day under short days), the middle stage (at 4 weeks under short days), and the late stage (at 7 weeks under short days). To standardize the plant phenophases, the study by [28] was adopted. The early stage indicates the initial transition stage for the reproductive growth (short day application) and corresponds to codes 37 to 39 in principal growth stage 3 (stem elongation). The middle stage corresponds to code 65 in principal growth stage 6 (flowering). Specifically, open flowering in female cannabis refers to the emergence of the stigma. The late stage marks the end of flowering, corresponding to the code 69 in principal growth stage 6. A more detailed graphical illustration of the phenophases for all experiments can be found in Appendix A.

### 2.3. Inflorescence and Flower Counting

Excluding the top apical meristem region, the number of inflorescences was counted as one per lateral branch, as shown in Figure 1. The top apical meristem region was difficult to distinguish accurately as it contained two or three inflorescences, so it was counted as one inflorescence. Several floral buds developed at the end of a single inflorescence, which can be classified as either female or male flowers depending on whether the stigmas or convex buds emerge, respectively. The stigmas of female flowers began to appear 3 weeks later under short day treatment, and after an additional 3 weeks, the fully developed female or male flowers were counted (see Figure 2 and Figure 3). Six weeks after female flowering, the stigmas began to turn brown and wither, as shown in Figure 2 and Figure 3. To identify male flowers in the middle reproductive stage, we counted flowers as male if they resembled those indicated by red arrows in Figure 2 and Figure 3. The number of buds in cluster 1 was a minimum of 12, in cluster 2 there were 9, and in cluster 3 a minimum of 6 buds were observed. Based on this criterion, the relative induction of male flowers was calculated by dividing the number of male flower buds formed by the number of buds observed for each cluster, expressed as a relative percentage based on the number of individuals.

### 2.4. Pollen Stainability Test

After 4 weeks in the short-day environment when plants were treated with STS once at the early productive stage, male buds started to open and anthers began to dehisce within 1 or 2 weeks, causing pollen release. At this stage, corresponding to codes 65 and 67 of the principal growth stage ‘flowering’ according to [28], the pollen grain stainability was tested. Three male flowers were randomly selected from three individuals. The three flowers were placed in a 1.5 mL eppendorf tube with 50% glycerol in deionized water, and vortexed for 30 s to release the pollen grains. After brief centrifugation for 20 s, the plant material on the solution surface was removed. The unstained pollen grains were mounted on a glass microscope slide, and viewed and photographed with an optical microscope (Leica). For potassium iodide–iodine (KI-I_2_) staining, 1% KI-I_2_ solution was added to the pollen in the tube (solution/water 1:1, *v*/*v*), lightly tapped to mix, then incubated at room temperature for 20 min. After brief centrifugation, the supernatant was discarded, then 200 μL of 50% glycerol was added and mixed by tapping. The sample was then observed and photographed under an optical microscope. Based on three observations, the number of stained and non-stained pollen grains were counted and the average calculated. In addition, pollen grains in 50% glycerol were stained with Alexander’s staining solution [22] for 30 min. The stained pollen grains were observed and photographed under a differential interference contrast microscope (Leica, Wetzlar, Germany).

### 2.5. STS Preparation and Treatment

Sodium thiosulfate (0.1 M; Duchefa, Haarlem, The Netherlands) and silver nitrate stock (0.1 M; Duchefa) were prepared in deionized water. To prepare 0.02 M STS solution, 20 mL silver nitrate was added to 80 mL sodium thiosulfate, and the solution was mixed to ensure that no clear white crystals formed with (silver/thiosulfate molar ratio = 1:4). The STS solution (20 mL) was sprayed onto an entire individual plant, as described in [16]. In addition, we applied STS as either one or three foliar sprays. The three treatments with STS spray were applied at 7-day intervals. The control plants were treated with water as a foliar spray coincident with STS application.

## 3. Results

### 3.1. Effects of STS Treatment at Early Reproductive Stage on Male Flower Induction

To investigate whether the induction ratio of male flowers was affected by the stage and frequency of STS treatment, the phenotype was analyzed at 6 weeks after treatment with one or three applications of STS at 7-day intervals. In the control group (0 mM STS, treated with water) for all three cultivars, no male flowers were induced (Figure 2A and Figure 3A), indicating that the three cultivars were perfect female plants. When the three cultivars were treated once with STS at the early reproductive stage, partial male flowers were induced in the 0.3 mM STS treatment for all cultivars observed at 6 weeks, although SuperwomanS1 showed greater induction of male flowers than the other two cultivars. In contrast, a high induction ratio of male flowers were observed in clusters 1 and 2 of Spectrum303 and SuperwomanS1 at 1.5 and 3.0 mM STS treatments (Figure 2A,B). CBGambit showed partial male flower induction in all STS concentrations, although 1.5 mM and 3.0 mM treatment of STS induced a higher frequency of male flowers than that in response to 0.3 mM STS (Figure 2B). After three STS applications at the early reproductive stage, partial male flowers of Spectrum303 and CBGambit were induced in the 0.3 mM STS treatment observed at 6 weeks, while three cultivars showed high male flower induction at 1.5 mM and 3.0 mM STS treatment (Figure 3A,B). SupwerwomanS1 treated with three STS applications showed a relatively higher induction ratio of male flowers than the other two cultivars and in response to a single STS application (Figure 2 and Figure 3). In the case of CBGambit, the increase in frequency of STS application resulted in more effective male flower induction (Figure 2 and Figure 3). These results indicated that the formation of male flowers was more effective under repetitive STS treatments, but also there may be a threshold STS concentration for induction of male flowers.
Figure 2Effects of a single STS application on cannabis plants at early and middle reproductive stages. (**A**) Features of the top inflorescence after STS treatment at three concentrations (0.3, 1.5, and 3.0 mM). Observation was performed at 6 weeks after STS treatment under short days. The red arrows indicate examples of male flowers. Black scale bars = 1 cm. (**B**,**C**) Percentage of individuals with male flowers at early stage (**B**) and middle stage (**C**). Numbers 1, 2, and 3 on the *X*-axis indicate cluster numbers. The different colors in histogram represent the relative percentage of male flowers in each plant (yellow, 0% male flower; gray, from 1% up to 50%; orange from 51% up to 99%, blue 100%). The *Y*-axis is the percentage of individuals with the relative percentage of male flowers. Thirty plants were used for each STS treatment (*n* = 30). Spe303, Spectrum303; SupS1, SuperwomanS1. (**D**,**E**) Numbers of inflorescence per plant height (cm) at early stage (**D**) and middle stage (**E**). Different lowercase letters in (**D**,**E**) denote statistical significance. Statistical analysis was performed using one-way analysis of variance with Tukey’s method and 95% confidence.
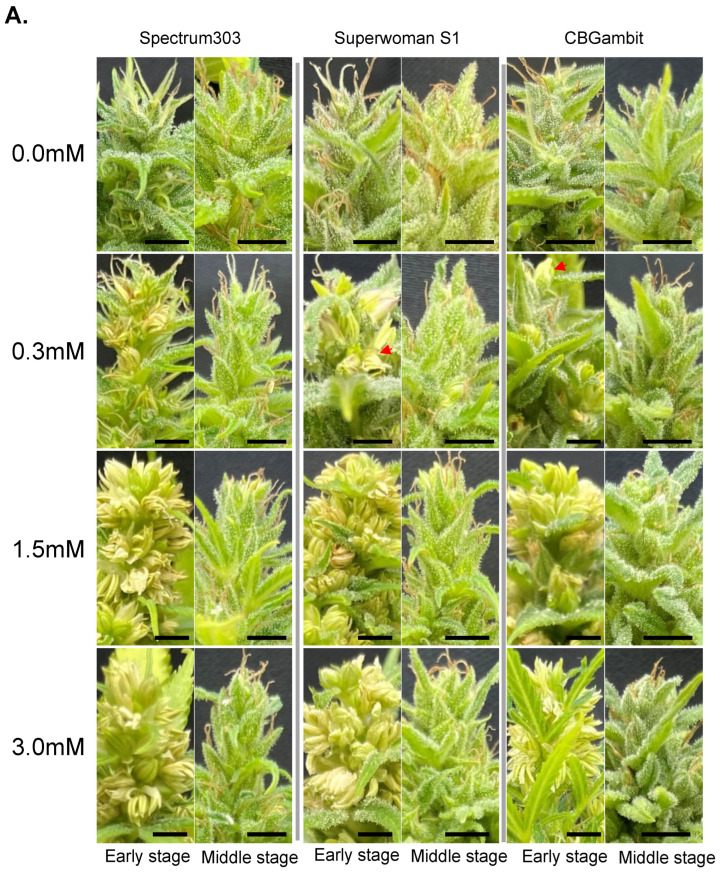

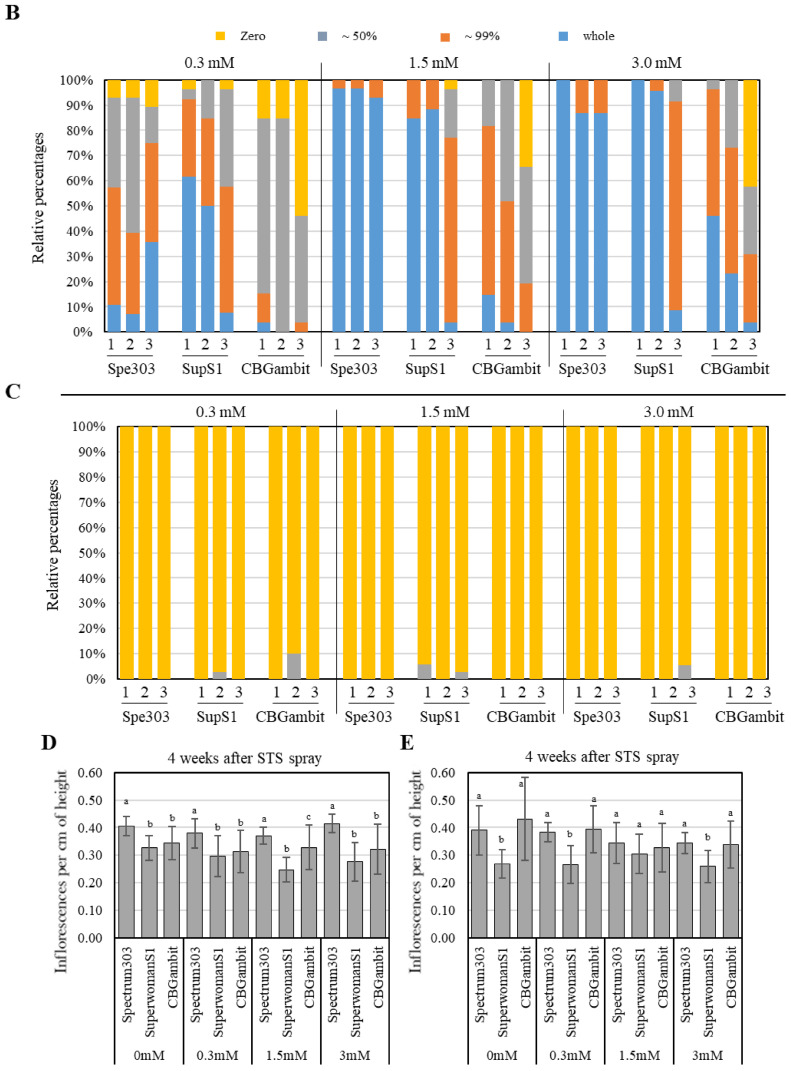



### 3.2. Effects of STS Treatment at Middle Reproductive Stage on Male Flower Induction

To develop a new cannabis cultivar, investigation of the optimal conditions for induction of male flowers on female cannabis plants is required. Previous studies have typically applied STS at the early reproductive stage. However, the cannabinoid contents should be analyzed at the mid- or late-flowering stages for the development of new cannabis cultivars for medicinal purposes, and following self- or cross-pollination. Therefore, we clarified whether feminized seeds can be obtained even after the full development of female flowers. When the three cultivars were treated once with STS at the middle reproductive stage, regardless of the STS concentration, the number of plants that produced a low frequency of male flowers (less than 50% male flowers, gray bars) comprised less than 10% of the total population at 6 weeks after treatment (Figure 2C). For breeding purposes, these flowers are considered to have a low potential to induce pollination in an individual plant. After three STS applications in the middle reproductive stage, although the number of male flowers per plant was low, the percentage of plants forming male flowers was highest, at approximately 57% in cluster 2 of Spectrum303 (Figure 3C). Moreover, CBGambit and SuperwomanS1 showed the highest relative ratio of male flower induction in cluster 1 by 53% of the plants at 0.3 mM STS and 26% at 1.5 mM STS, respectively (Figure 3C). We speculated that the lower frequency of male flower induction was due to sporadic formation of male flowers from the late-developing germ cells affected by STS. These results indicated that induction of male flowers at the middle reproductive stage, which is required for the seeds set through self- or cross-pollination, was possible with repetitive STS treatments.

### 3.3. Effects of Concentration of STS Treatment on Male Flower Induction

To investigate whether the induction ratio of male flowers was dependent on the STS concentration, three concentrations (0.3, 1.5, and 3.0 mM) were applied to the three cannabis cultivars. In Spectrum303 with single STS application, the approximately 10% proportion of whole male flowers produced in response to 0.3 mM STS was increased to more than 90% in response to 1.5 and 3.0 mM STS. This pattern of male flower induction was similarly observed in SuperwomanS1 and CBGambit, indicating that the number of individuals producing whole male flowers increased gradually with increase in the STS concentration (Figure 2B). However, no significant difference in the ratio of male flower induction of Spectrum303 and SuperwomanS1 was observed between 1.5 and 3.0 mM STS concentrations (Figure 2B). The induction of male flowers in cluster 3 did not increase dramatically with increase in STS concentration, unlike the response observed in clusters 1 and 2 (Figure 2B).

In the case of three applications of STS, Spectrum303 showed partial male flower induction in response to 0.3 mM STS, whereas a strong increase in male flower induction was observed in response to 1.5 and 3.0 mM STS treatment (Figure 3B). All cases of SuperwomanS1 induced mostly whole male flowers in response to three STS applications for all concentrations (Figure 3B). In CBGambit with three applications of STS, approximately 60% of cases produced whole male flowers in response to 0.3 mM STS, which increased to 80% at 1.5 mM STS and 100% at 3.0 mM STS (Figure 3B). Thus, in the cluster 1 and 2 regions, the increase in STS concentration led to an increase in induction ratio of male flowers in all three cultivars up to the highest concentration of STS.
Figure 3Effects of three STS applications on cannabis plants at early and middle reproductive stages. (**A**) Features of the top inflorescence after three STS applications at three concentrations (0.3, 1.5, and 3.0 mM). Observation was performed at 6 weeks after STS treatment. The red arrows indicate examples of male flowers. Black scale bars = 1 cm. (**B**,**C**) Percentage of individuals with male flowers at early stage (**B**) and middle stage (**C**). Numbers 1, 2, and 3 on *X*-axis indicate cluster numbers. The different colors in histogram represent the relative percentage of male flowers in each plant (yellow, 0% male flower; gray, from 1% up to 50%; orange, from 51% up to 99%, blue, 100%). The *Y*-axis is the percentage of individuals with relative percentage of male flowers. Thirty-six SuperwomanS1 plants (*n* = 36), and 12 Sperctrum303 and CBGambit plants (*n* = 12) were used for each STS treatment. Spe303, Spectrum303; SupS1, SuperwomanS1. (**D**,**E**) Numbers of inflorescences per plant height (cm) at early stage (**D**) and middle stage (**E**). Different lowercase letters in (**D**,**E**) denote statistical significance. Statistical analysis was performed using one-way analysis of variance with Tukey’s method and 95% confidence.
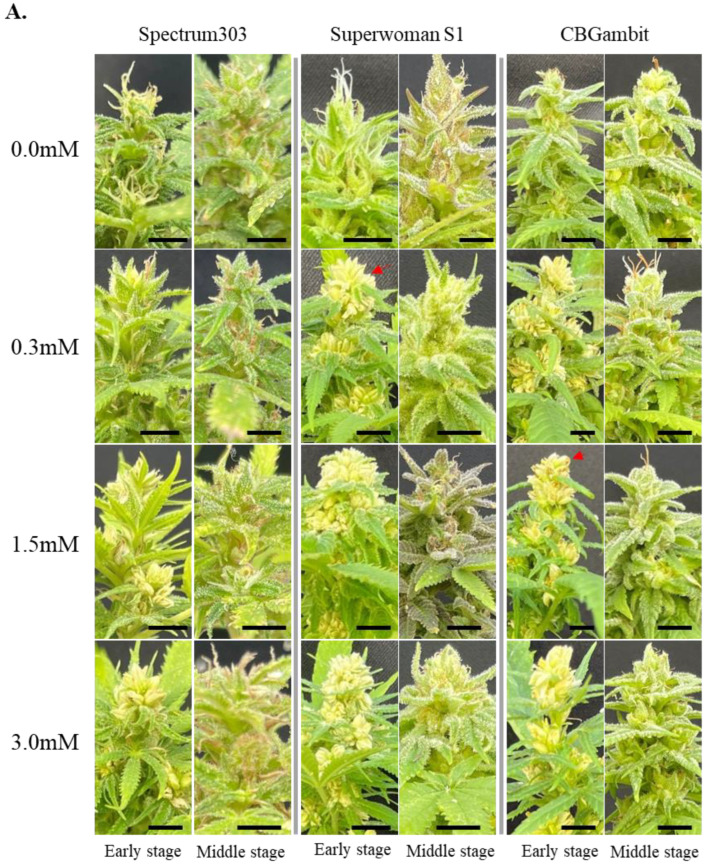

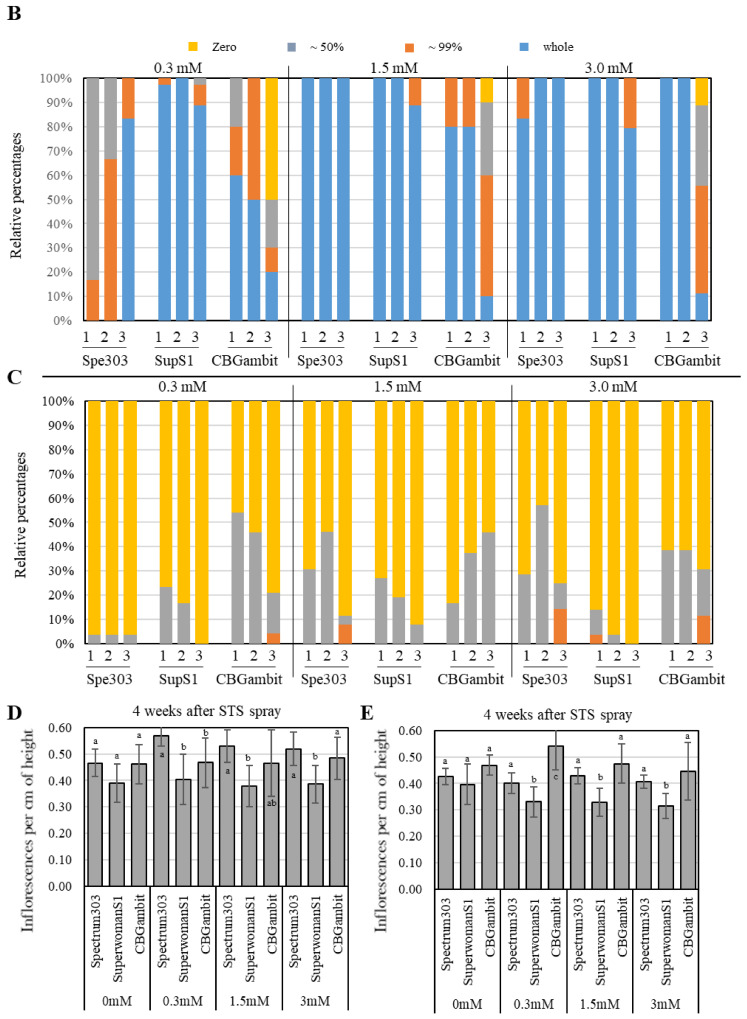



### 3.4. Impacts of STS Treatment on Inflorescence Development at Two Reproductive Stages

To investigate how the number of inflorescences was affected by the timing, frequency, and concentration of STS treatment, the number of inflorescences and shoot lengths were measured in the three cultivars at 4 weeks after STS treatment at the early and middle reproductive stages. After a single STS application at both stages, SuperwomanS1 plants had relatively fewer inflorescences per shoot compared with other cultivars (Figure 2D,E). This result was consistent with the response to three STS applications (Figure 3D,E).

In the early stage following a single STS application, the mean number of inflorescences per centimeter (cm) of shoot length was approximately 0.41 for Spectrum303, 0.31 for SuperwomanS1, and 0.35 for CBGambit (Figure 2D). In the middle stage following a single STS application, the mean number of inflorescences per cm of shoot length was approximately 0.38 for Spectrum303, 0.27 for SuperwomanS1, and 0.37 for CBGambit (Figure 2E). These values were not significantly affected by the increase in STS concentration.

Following three STS applications, the mean number of inflorescences per cm of shoot length in the early stage was approximately 0.51 for Spectrum303, 0.39 for SuperwomanS1, and 0.46 for CBGambit (Figure 3D). In the middle stage, the mean number of inflorescences per cm of shoot length was approximately 0.43 for Spectrum303, 0.34 for SuperwomanS1, and 0.48 for CBGambit (Figure 2E). Similar to the response to a single STS application, the number of inflorescences was not influenced by the increase in STS concentration. These results indicated that the number of inflorescences per shoot length was not affected by STS concentration and application frequency in both the early and middle reproductive stages.

### 3.5. Pollen Stainability in Male Flowers of the Three Cultivars

To investigate whether seed formation was possible using pollen grains from male flowers produced in response to STS treatment, we examined the stainability of the pollen grains as an indicator of their viability. For this purpose, pollen grains from male flowers of plants of the three cultivars treated with a single STS application at different concentrations at the early reproductive stage were stained with KI-I_2_ solution and Alexander’s staining solution. The KI-I_2_ solution indicates the presence or absence of starch, a nutrient required for pollen development [23,29]. First, we observed the shape of the non-stained pollen grains in 50% glycerol solution. The pollen grains were round, similar to those produced by other *Cannabis* species (Figure 4B). Pollen grains from male flowers induced by different STS concentrations at the early stage appeared yellow at 10× magnification and purple at 40× magnification after staining with KI-I_2_ solution (Figure 4A). Non-stained pollen grains at 40× magnification comprised approximately 22%, 8.2%, and 8.3% of the pollen in Spectrum303, SuperwomanS1, and CBGambit, respectively, in the 0.3 mM STS treatment. The percentage of stained pollen grains increased in Spectrum303 and CBGambit with increases in STS concentration (Figure 4). In SuperwomanS1, pollen stainability was higher in the 3.0 mM STS treatment than in response to the other two concentrations (Figure 4C). Overall, higher STS concentrations showed a tendency for improved pollen stainability.

Given that the KI-I_2_ staining method is less capable of distinguishing between viable and non-viable pollen grains, we used Alexander’s staining method to confirm the results [30]. Alexander’s staining solution distinguishes live and dead cells based on cell pH, with red coloration indicating live cells and blue coloration indicating dead cells [22,31]. Non-viable pollen grains comprised approximately 34%, 20%, and 22% of the pollen grains in the 0.3, 1.5, and 3.0 mM STS treatments for Spectrum303 (Appendix A). In the case of SuperwomanS1, the percentage of non-viable pollen grains remained similar regardless of the STS concentration. Similar to KI-I_2_ staining, in Spectrum303 improved pollen viability was observed in response to 1.5 or 3.0 mM STS compared with that in the 0.3 mM STS treatment. Interestingly, the pollen viability of the two cultivars was decreased compared with KI-I_2_ staining results. The reason is likely to be the differences in ambient temperature and method. In the KI-I_2_ experiment, the average daytime temperature was 30 °C, whereas in the Alexander’s staining experiment, owing to the difference in season, the average daytime temperature was 37 °C.

## 4. Discussion

In this study, we investigated the effects of different concentrations and application frequencies of STS on male flower induction in three female cannabis cultivars at the early and middle reproductive stages under short days. Regarding SuperwomanS1, a sufficient number of male flowers were obtained in response to 0.3 mM STS treatment, whereas for Spectrum303 and CBGambit a concentration of 1.5 mM STS or higher was required when applied at the early stage. To ensure the development of an adequate number of male flowers on female cannabis plants, it is advisable to apply STS during the early stage under short days. Moreover, increasing the STS application frequency during the middle stage may be beneficial for the breeding of individual plants. Pollen in male flowers induced by foliar application of STS exhibited relatively high viability regardless of the STS concentration, indicating there is no obstacle to obtaining feminized seeds. Considering these results, adjusting the timing and frequency of STS treatments according to the intended purpose would be beneficial for effective cannabis breeding and feminized seed production.

### 4.1. Influence of STS Treatment on Stem Height and Inflorescence Number

The STS is a more effective chemical than silver nitrate (AgNO_3_) or gibberellin for induction of male flowers in female cannabis plants [16,20,32]. The foliar-spray method of STS application, as mentioned in the study by [16], does not cause severe plant injury when a large volume is directly applied to the shoot tip. In the present study, the foliar-spray application of STS did not result in significant differences in the number of stems and inflorescences compared with the water-treated group (the control), regardless of the application frequency or concentration, as observed in all three cultivars (Figure 2 and Figure 3). These results are consistent with previous findings in which no significant differences in the number of stems and nodes from the control group were detected after three foliar-spray applications of 20 and 0.3 mM STS [20].

### 4.2. Recommended Concentration and Frequency of STS Application for Specific Purposes

The reason for STS application is to obtain feminized seeds with two X chromosomes because female cannabis flowers produce (or accumulate) more cannabinoids compared with male cannabis flowers [33,34]. It is logical to induce a reasonable number of male flowers to pollinate female flowers for bulk production of feminized seeds. High concentrations tend to result in improved male flower induction [16,35]. Therefore, application of a sufficiently high concentration of STS is required to obtain an adequate number of male flowers. In this study, we observed a notable formation of male flowers in response to treatment with an STS concentration of 1.5 mM or higher, regardless of the application frequency, in Spectrum303 and SuperwomanS1 at the early reproductive stage (Figure 2 and Figure 3). Interestingly, even with a single application of 0.3 mM STS, more than 90% of SuperwomanS1 plants produced exclusively male flowers, whereas Spectrum303 and CBGambit produced lower frequencies of male flowers (Figure 2). This difference is attributable to variation in the response to STS among cannabis cultivars. Similarly, in the study by [16], CBD hemp A was more responsive to 0.3 mM STS treatment compared with three other hemp varieties, resulting in a higher frequency of male flowers. Furthermore, after treatment of MX-CBD-707 hemp with a single application of 0.7 mM STS, no significant difference was observed in the number of male flowers compared with treatment of the same variety with 20 mM STS [20]. Based on the present and previous studies, it can be concluded that, although it may vary among cultivars, the production of exclusively male flowers in cannabis individuals is achievable by a single treatment with STS at a specific threshold or higher concentration.

Previous studies that examined the correlation between STS application and male flower induction applied short days after STS treatment. When subjected to the short days, cannabis plants undergo the transition from vegetative growth to reproductive growth, and the mother cells of female flowers gradually develop into male flowers in response to STS [19,36]. In addition, to obtain many male flowers on a female cannabis plant, the frequency of STS application is not a major concern, provided that the STS concentration is maintained at a certain level. Therefore, STS treatment during the middle reproductive stage when female flowers are nearing maturity may be unnecessary. Following a single STS application during the middle reproductive stage, only a small number of male flowers developed (Figure 3C), suggesting that STS applications at the middle stage may not be beneficial. However, the present study revealed one potential utilization of treatment with three STS applications during the middle reproductive stage. Three applications of STS resulted in a significant increase in the number of individual plants with male flowers (although the number of male flowers per individual plant was low), which comprised more than 55% in cluster 2 of Spectrum303 and approximately 54% in cluster 1 of CBGambit (Figure 3C). These results implied that, as the frequency of STS applications increases, the conversion of reproductive cells formed late in the reproductive stage into male flowers gradually increases. From the perspective of cannabis breeding, this treatment may be advantageous for cannabinoid analysis in unfertilized female flowers and subsequent induction of male flowers for pollination. Nevertheless, additional studies of the discrepancy of pollination timing and probability of successful fertilization during the middle or late stage of reproductive growth are required.

### 4.3. Seed Production by Pollen Viability

Previous studies have confirmed that female cannabis plants can form viable pollen in male flowers and produce seeds [16]. However, how the viability of cannabis pollen varies with different concentrations of STS is poorly understood. Although [20] evaluated pollen viability using fluorescein diacetate, a statistical analysis was not performed. Based on the present results, which indicated more than 80% pollen viability in all three cultivars, it can be inferred that harvesting seeds is unlikely to be a challenge (Figure 4). This finding is supported by previous experiments that demonstrated adequate pollen germination for seed production, as indicated by in vitro and in vivo pollen germination assays, despite relatively low percentages of germinated pollen grains [20,27]. Approximately 5% of the pollen of CBGambit was non-viable (Figure 4C). It is possible that some pollen grains may have been damaged during experimental procedures, such as vortexing and centrifugation for pollen staining. However, it can be assumed that the majority of the pollen grains are viable and capable of effecting fertilization. Notably, sufficient seeds were harvested from individual female plants that were pollinated with pollen grains from male flowers induced by a single STS application at the early reproductive stage. In the case of three application of STS, moreover, pollen viability was not significantly different from the results of a single STS application.

Pollen viability is affected by high temperature and relative humidity [37,38]. Alexander’s staining indicated a lower percentage of viable pollen grains compared with the results of KI-I_2_ staining. Pollen subjected to the stress from a high daytime temperature may exhibit a higher frequency of pollen abortion. However, a previous study suggests that pollen viability assessed by histochemical staining methods in cannabis may be inconclusive, because not all non-aborted pollen grains are viable [39]. In vitro pollen germination assay is an important method for assessing pollen viability. Previous studies on in vitro pollen germination have reported that the germination rate of cannabis pollen from male plants under various culture media conditions ranged from 10% to 84% [24,25,26,27]. Additionally, the germination rate of pollen grains induced by STS under in vitro conditions was reported to be up to 13% [27] and less than 10% [40]. These findings indicate that an appropriate pollen germination assay for cannabis has not yet been established. After optimizing the in vitro germination conditions for cannabis pollen grains, further research is necessary to investigate the precise pollen viability in male flowers induced by STS.

### 4.4. Potential Applications

Our report presents useful information regarding producing male flowers on female cannabis plants for various intended uses. If bulk feminized seeds are required, apply a single STS application at the early reproductive stage to induce conversion of full male flowers. Moreover, a threshold response to STS was observed among different cultivars. If you use SuperwomanS1, which can develop many male flowers even when treated with a low concentration of 0.3 mM STS (Figure 2), you can minimize chemical use for ready conversion to male flowers. If feminized seeds are required for cannabis breeding, apply three STS applications at the middle reproductive stage to enable self-pollination after cannabinoid analysis.

Female cannabis plants are valuable for the cultivation and production of cannabinoids. To feminize the cannabis seeds, STS treatment on female plants is widely used. This study examined the effects of STS concentration, frequency, and application timing, providing insights on how researchers or growers can apply STS to achieve their desired goals. Moreover, viability tests of pollen induced by STS have shown sufficient pollination potential, regardless of STS concentration. This information will benefit breeding programs and the production of feminized seeds.

## Figures and Tables

**Figure 1 plants-13-02429-f001:**
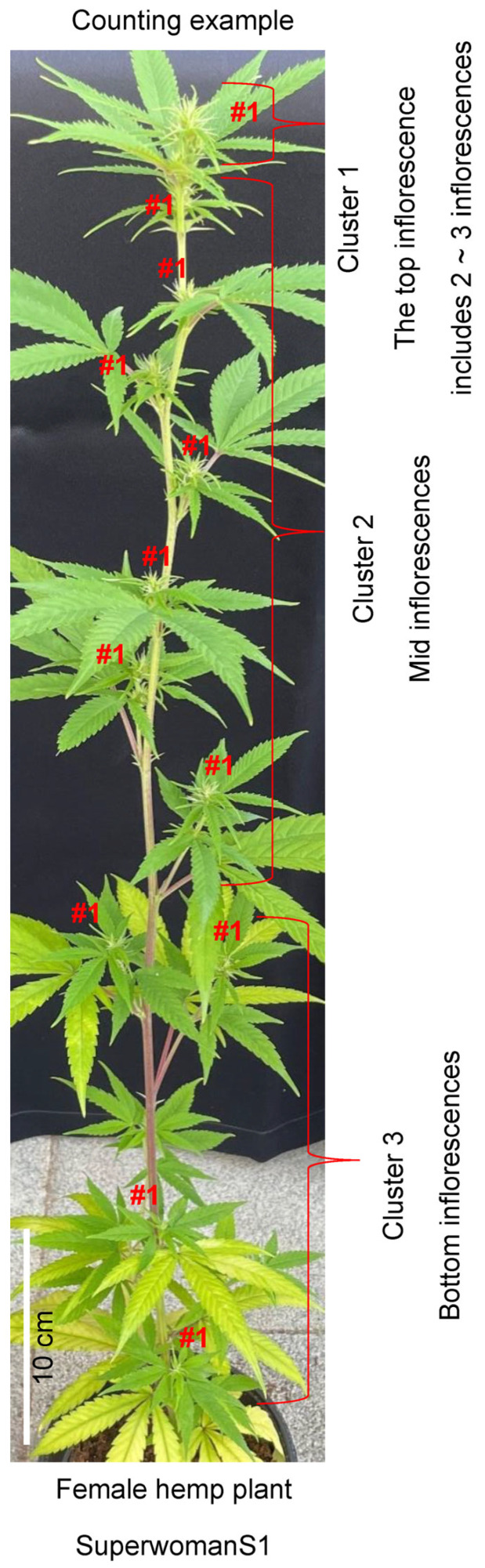
Example of a female cannabis (*Cannabis sativa*) plant for measuring the number of inflorescences and male flowers. To measure the number of inflorescences, “#1” symbols indicate the counted number of inflorescences that developed based on leaf arrangement, including the top and the auxiliary stems. Among these, three clusters (1, 2, and 3; top, middle, and bottom) were distinguished. One randomly selected inflorescence from each cluster was used to count the number of male flowers. Scale bar = 10 cm.

**Figure 4 plants-13-02429-f004:**
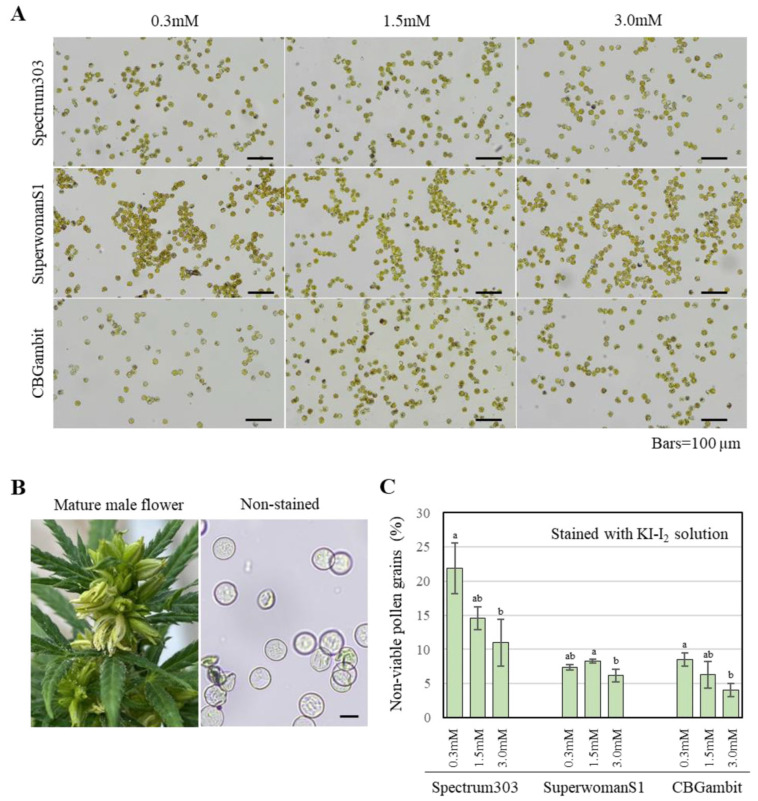
Pollen stainability from fully matured male flowers on female cannabis treated with a single STS treatment. (**A**) Pollen grains stained with KI-I_2_ solution. Bars = 100 µm. (**B**) Mature male flower and pollen grains. Image of non-stained pollen grains in 50% glycerol was captured under a light-microscope. Bar = 50 µm. (**C**) Pollen grain stainability with KI-I_2_ solution. Staining was performed with three biological replicates. Different lowercase letters denote statistical significance. Statistical analysis was performed using one-way analysis of variance with Tukey’s method and 95% confidence.

## Data Availability

The original contributions presented in the study are included in the article/Appendix A, further inquiries can be directed to the corresponding author.

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
