# Peer review of "Characterization of Male Flower Induction by Silver Thiosulfate Foliar Spray in Female Cannabis at the Middle Reproductive Stage for Breeding"

_plants, 2024, doi:10.3390/plants13172429_

Round 1

Reviewer 1 Report

Comments and Suggestions for Authors

This work focused on the male flower induction from the female individules by the foliar spray of STS. However, there were some issues need to be modified or improved. 1. The introduction was too long, and however, most sentences were centered on the social and economic values, not on the scientific significance. Also the research progress should be stated in the Introduction part. For instance, whether the female individual could generate male chromsome, the fertility of sperm, and whether the fertilized offspring remained female? This will be more interesting for the production. 2. the authors also assessedthe pollen viability, but were not mentioned in the abstract.

Author Response

Comment 1: This work focused on the male flower induction from the female individuals by the foliar spray of STS. However, there were some issues need to be modified or improved. 
Response 1: Thank you for your critical review. We tried to modify or improve the manuscript following your suggestions.

Comment 2: The introduction was too long, and however, most sentences were centered on the social and economic values, not on the scientific significance. Also the research progress should be stated in the Introduction part. For instance, whether the female individual could generate male chromsome, the fertility of sperm, and whether the fertilized offspring remained female? This will be more interesting for the production. 
Response 2: Thank you for your valuable suggestion. We tried to revise some irrelevant sentences on our research focus from the introduction. The following shows the sentences that were removed or inserted in the introduction part.
Deletion: 
1. At line 53 to 55 of original MS, “The distinction between these two terms also reflects a difference in THC concentration, which is a psychoactive substance classified as a hallucinogen.” 
2. At line 67 to 72 of original MS, “Of these compounds, CBD has no psychoactive effects in contrast to THC, and is used as a medical treatment. Given its potential uses, cannabis cultivation for medical or recreational purposes has been legalized through licensing in 21 states in the United States (Conway 2023; www.statista.com). As a result, the area of cannabis cultivation in the United States has continued to increase, with a total of 21,853 hm2 reported by the United States Department of Agriculture (USDA) as of 2021.” 
3. At line 84 to 95 of original MS, “In particular, the market value of CBD, a recently emergent substance in the pharmaceutical market, is expected to reach approximately US$16 billion by 2025 (Dorbian, 2019). For recreational use, numerous cultivars with high THC contents are not well documented because they are implicitly cultivated. Cultivars that produce a psychoactive effect often have additional terpenes that elicit a unique aroma and are frequently cultivated (Peters and Chien et al., 2018). THC is a terpene-based substance, so locating a mutation or recombination in the genes responsible for the synthesis of other terpenes in the same biosynthetic pathway that produce a unique aroma is relevant to both recreational and medical uses (Kovalchuk et al., 2020). Terpenes give the plant its characteristic taste and smell. These molecules in cannabis contribute to the consumer attraction and market price, and enhance many therapeutic benefits, especially in aromatherapy (Sommano et al., 2020; Hanuš and Hod, 2020). Therefore, recreational and medicinal applications should not be considered separately.”
4. At line 104 to 110 of original MS, “Growers prefer cultivating female plants for medicinal purposes because they tend to accumulate higher contents of cannabinoids than male plants (Rahn et al., 2016; Soler et al., 2017). To develop new varieties of cannabis for the aforementioned purposes, various biological techniques, including traditional cross- and mutation- breeding methods, can be used. In some instances, techniques such as tissue culture are used for in vitro biomolecule synthesis. Genetic engineering in combination with CRISPR-CAS9 gene editing is used to generate cultivars with new genetic traits (Adhikary et al., 2021; Hesami et al., 2021). Generally, landrace cannabis seeds have XX or XY chromosomes with a 1:1 ratio.”
Insertions: 
1. At line 81 to 84 of the revised MS, “In order to achieve the manipulation of cannabinoids, the need to continuously maintain female plants has arisen, since growers prefer cultivating female plants for medicinal purposes because they tend to accumulate higher contents of cannabinoids than male plants (Rahn et al., 2016; Soler et al., 2017).”
2. At line 110 to 117 of the revised MS, “The viability of pollen can be assessed by staining methods to check its vitality or by examining pollen germination (Bengtsson, 2006). Representative examples of staining methods include KI-I2 solution and Alexander’s staining (Alexander, 1969; Melloni et al., 2013). Viability tests of cannabis pollen have primarily been conducted using in vitro pollen germination assay on pollen grains obtained from male plants (Zottini et al., 1997; Gaudet et al., 2020; Wizenberg et al., 2022). Studies have also reported that pollen from male flowers induced by chemical inducers, such as STS, is viable both in vivo and in vitro. (Dimatteo et al., 2020; Flajšman et al., 2021). However, studies on the changes in pollen viability depending on the concentration of STS are still limited.”
Move:
1. At line 124 to 126 from introduction part, a sentence “High concentrations tend to result in improved male flower induction (Moon et al., 2010; Lubell and Brand, 2018).” was removed to the discussion part, at line 347 to 348 of the revised MS.

Comment 3: The authors also assessed the pollen viability, but were not mentioned in the abstract.
Response 3: You are right. Thank you for pointing that out, and we have revised this point to the Abstract. At line 36 to 38 of the revised MS, “Pollen stainability tests using KI-I2 solution and Alexander’s staining showed high pollen viability with over 65% at different single STS concentrations, indicating that pollen grains induced by STS have sufficient viability for the self-pollination.” was added.

Reviewer 2 Report

Comments and Suggestions for Authors

The STS method is commonly used in cannabis breeding and production to induce male flowers in female plants.  This article provides a detailed study on the induction of male flowers by STS, which has certain scientific significance and references.

However, I believe that the article lacks an important data point, which is the comparison between the quantity and germination pollination ability of male pollen induced by STS and normal male pollen.  In addition, germination rate of pollen on the culture medium will be more convinced instead of the KI-I2 staining.

It will more intuitively reflect the scientific value of this article if the authors consider the above suggestions.

Comments on the Quality of English Language

I have no comments on the Quality of English Language.

Author Response

Comment 1: The STS method is commonly used in cannabis breeding and production to induce male flowers in female plants.  This article provides a detailed study on the induction of male flowers by STS, which has certain scientific significance and references.
Response 1: Thank you for your critical review. We tried to provide precise details regarding the use of the chemical STS, including its frequency, concentration, and application timing to induce male flowers on female plants.

Comment 2: However, I believe that the article lacks an important data point, which is the comparison between the quantity and germination pollination ability of male pollen induced by STS and normal male pollen. In addition, germination rate of pollen on the culture medium will be more convinced instead of the KI-I2 staining. It will more intuitively reflect the scientific value of this article if the authors consider the above suggestions.
Response 2: Thank you for the valuable suggestion. Conducting a pollen germination assay would indeed generate significant data. Initially, we considered an in vitro pollen germination assay for pollen induced by STS, given the inconsistencies between pollen viability and non-abortion rate (or germination rate) reported in the literature (Wizenberg et al., 2022). However, our review of the existing studies on in vitro pollen germination, whether from STS-induced pollen or normal male plants, revealed significant variability in germination rates. For example, in male plants, the in vitro germination rate ranged from 10% to 84% depending on the culture medium. Specifically, Zottini et al. (1997) reported 84%, Wizenberg et al. (2022) reported approximately 40%, Gaudet et al. (2020) recorded about 50%, and Dimatteo et al. (2020) reported 10%. Moreover, for pollen grains induced by STS, the in vitro germination rate was reported to be between 0-13% (Dimatteo et al., 2020) and less than 10% (Fitzgerald, 2020). This led us to conclude that an optimal in vitro pollen germination assay for cannabis pollen, whether induced by STS or from male plants, has not yet been properly established. As a result, we focused on assessing pollen viability using two histochemical assays (KI-I2 solution and Alexander staining). Therefore, we believe that optimizing the in vitro germination conditions for hemp pollen grains induced by STS or from male plants should be prioritized before comparing the quantity and germination ability of pollen induced by STS. This and the viability comparison between normal male pollen and the induced pollen will be our next challenge.
We have included this discussion at line 406 to 414 of the Discussion Section of the revised MS:
“In vitro pollen germination assay is an important method for assessing pollen viability. Previous studies on in vitro pollen germination have reported that the germination rate of cannabis pollen from male plants under various culture media conditions ranged from 10% to 84% (Zottini et al., 1997; Dimatteo et al., 2020; Gaudet et al., 2020; Wizenberg et al., 2022). Additionally, the germination rate of pollen grains induced by STS under in vitro conditions was reported to be up to 13% (Dimatteo et al., 2020) and less than 10% (Fitzgerald, 2020). These findings indicate that an appropriate pollen germination assay for cannabis has not yet been established. After optimizing the in vitro germination conditions for cannabis pollen grains, further research is necessary to investigate the precise pollen viability in male flowers induced by STS.”

Reviewer 3 Report

Comments and Suggestions for Authors

The manuscript with the title “Characterization of male flower induction by silver thiosulfate foliar spray in female cannabis at the middle reproductive stage for breeding” investigated male flower induction in three commercial cultivars of female cannabis using several silver thiosulfate concentrations.

The Introduction provides the necessary background and documentation for the research.

Material and Method, Results

I suggest phenophases to always be defined as well based on a standardized system largely in use (at least giving the code), such as BBCH system or other system. Just naming them Early and Middle stage is not sufficiently precise. Also, for example in chapter 2.3. as well, transition from a secondary phenophase to another one can be explained using standardized terminology. I think there are published papers on flowering phenology of hemp that might be helpful here, or if not, a table in Material and Method could clarify principal and secondary stages of flowering observed here and delimit them precisely (section 2.2). Still, I suggest using a standardized phenology scale of choice. This will help future authors to compare their work with yours, increasing the chances of your paper to be cited.  

Figure 2 – what the whiskers represent (±SD, SE, CI…) please insert the explanation in the figure caption.

Given that the manuscript has no Conclusions section, only the abstract provides a summary of the research.

Best regards.

Author Response

Comment 1: The manuscript with the title “Characterization of male flower induction by silver thiosulfate foliar spray in female cannabis at the middle reproductive stage for breeding” investigated male flower induction in three commercial cultivars of female cannabis using several silver thiosulfate concentrations.
Response 1: Yes, right. Thank you for your critical reviews. 

Comment 2: The Introduction provides the necessary background and documentation for the research.
Response 2: Thank you for your kind compliment. One of the reviewers requested improvements to the Introduction section, specifically to reduce the content related to social and economic values, so it has been revised. You can see it from the revised MS.

Comment 3: Material and Method, Results
I suggest phenophases to always be defined as well based on a standardized system largely in use (at least giving the code), such as BBCH system or other system. Just naming them Early and Middle stage is not sufficiently precise. Also, for example in chapter 2.3. as well, transition from a secondary phenophase to another one can be explained using standardized terminology. I think there are published papers on flowering phenology of hemp that might be helpful here, or if not, a table in Material and Method could clarify principal and secondary stages of flowering observed here and delimit them precisely (section 2.2). Still, I suggest using a standardized phenology scale of choice. This will help future authors to compare their work with yours, increasing the chances of your paper to be cited.  
Response 3: Thank you for your valuable suggestion. Mishchenko et al (2017) provided a phenological growth stage of hemp with BBCH system. As you know, the female and male cannabis plant have different feature of flowers. Mishchenko study’s footnote indicated about the difficulties in allocation of female flower development at the code. Regarding this, however, we can give the code according to above paper and describe the stage as an image in Supplementary Figure 2 to avoid misunderstanding. This will be more helpful for the readers and future authors. The following context below is the revised sentence or section.
At line 149 to 158 in the revised MS (at material and method part), we have added more information about above suggestion; 
“Phenological growth stage of female cannabis used in this study
We divided reproductive development into three stages, namely, the early stage (from the first day under short days), the middle stage (at 4 weeks under short days), and the late stage (at 7 weeks under short days). To standardize the plant phenophases, the study by Mishchenko et al (2017) was adopted. The early stage indicates the initial transition stage for the reproductive growth (short day application) and corresponds to the code 37 to 39 in principal growth stage 3 (stem elongation). The middle stage corresponds to the code 65 in principal growth stage 6 (flowering). Specifically, open flowering in female cannabis refers to the emergence of the stigma. The late stage marks the end of flowering corresponding to the code 69 in principal growth stage 6. A more detailed graphical illustration of the phenophases for all experiments can be found in supplementary Figure S2.”
At line 178 to 179 in the revised MS (at material and method part), “At this stage, corresponding to codes 65 and 67 of the principal growth stage ‘flowering’ according to Mishchenko et al (2017),….” was added.

Comment 4: Figure 2 – what the whiskers represent (±SD, SE, CI…) please insert the explanation in the figure caption.
Response 4 : Thank you for your pointing to Figure 2 caption. We provided the figure captions in PPT slide. Figure captions now also can be found in the revised MS. In Figure 2, we represented a data as percentage of individuals. It means that if number of individuals from n=30 with 0% male flower is 30 ea, it will be 100%. Given that one experiment was conducted with a sample size of n=30, SD cannot be calculated.

Comment 5: Given that the manuscript has no Conclusions section, only the abstract provides a summary of the research.
Response 5: Thank you for your pointing to the conclusion. We have added a conclusion remarks in end of the discussion part. At line 425 to 430 in the revised MS, “Female cannabis plants are valuable for the cultivation and production of cannabinoids. To feminize the cannabis seeds, STS treatment on female plants is widely used. This study examined the effects of STS concentration, frequency, and application timing, providing insights on how researchers or growers can apply STS to achieve their desired goals. Moreover, viability tests of pollen induced by STS have shown sufficient pollination potential, regardless of STS concentration. This information will benefit breeding programs and the production of feminized seeds.”

Round 2

Reviewer 3 Report

Comments and Suggestions for Authors

Dear authors,

the comments were addressed and issues were fixed.

Best regards.